

# cdev: a ground-truth based measure to evaluate RNA-seq normalization performance

Diem-Trang Tran[1] and Matthew Might[2]

[1] School of Computing, University of Utah, Salt Lake City, UT, United States of America
[2] Hugh Kaul Precision Medicine Institute, University of Alabama at Birmingham, Birmingham, AL, United States of America

## ABSTRACT

Normalization of RNA-seq data has been an active area of research since the problem was first recognized a decade ago. Despite the active development of new normalizers, their performance measures have been given little attention. To evaluate normalizers, researchers have been relying on *ad hoc* measures, most of which are either qualitative, potentially biased, or easily confounded by parametric choices of downstream analysis. We propose a metric called condition-number based deviation, or *cdev,* to quantify normalization success. *cdev* measures how much an expression matrix differs from another. If a ground truth normalization is given, *cdev* can then be used to evaluate the performance of normalizers. To establish experimental ground truth, we compiled an extensive set of public RNA-seq assays with external spike-ins. This data collection, together with *cdev,* provides a valuable toolset for benchmarking new and existing normalization methods.

## INTRODUCTION

Between-sample normalization has long been recognized as a critical step in processing and analyzing RNA-sequencing (RNA-seq) data. The problem has been studied for a decade, resulting in a large array of methods (*Bullard et al., 2010*; *Robinson & Oshlack, 2010*; *Anders & Huber, 2010*; *Kadota, Nishiyama & Shimizu, 2012*; *Li et al., 2012*; *Glusman et al., 2013*; *Maza et al., 2013*; *Chen et al., 2014*; *Zhuo et al., 2016*; *Roca et al., 2017*; *Tran et al., 2020*; *Dos Santos, Desgagné-Penix & Germain, 2020*; *Wang, 2020*). Despite such active development of new normalizers, evaluation schemes have mostly been impromptu. The performance of normalizers has been measured either directly on their output, *i.e.,* the normalized expression levels, or indirectly *via* the results of downstream analyses, such as differential expression (DE) detection, classification, or clustering. As a prominent application of RNA-seq data, DE analysis has been routinely used to assess normalization quality (*Bullard et al., 2010*; *Dillies et al., 2013*; *Maza et al., 2013*; *Peixoto et al., 2015*; *Evans, Hardin & Stoebel, 2017*). Picking a DE calling method from a plethora of options (*Soneson & Delorenzi, 2013*; *Tang et al., 2015*) is mostly arbitrary, yet may introduce unknown biases. Clustering is another common indicator of normalization performance

Corresponding author
Diem-Trang Tran,
dtrang.tran@utah.edu

(*Rapaport et al., 2013*; *Risso et al., 2014a*). Intuitively, the clustering of successfully normalized expression profiles should reflect natural and meaningful partition of the samples, such as tissue types, source organisms, cancer subtypes, etc. The recovery of such partitions can be measured by rand index, Jaccard similarity, etc., which then serve as the proxies for normalization performance. Except for clustering based on principal component analysis (PCA) which can be easily reproduced thanks to the uniqueness of the singular value decomposition, most clustering analyses are the result of multi-faceted decision regarding the selection and transformation of features, the type of distance measure, the clustering algorithm and its parameters. Eventually, such analysis may be too low in resolution to differentiate normalization methods (*Rapaport et al., 2013*). Other machine learning applications such as classification and clinical outcome prediction have also been used to evaluate the quality of RNA-seq processing workflow, including normalization (*Zyprych-Walczak et al., 2015*; *Tong et al., 2020*). Similar to clustering, these applications involve hyper-parametric choices such as the type of feature transformation, the feature selection procedure, the type of classifier, etc. all of which effectively create an unlimited number of workflows, each of which may lead to a different ranking of normalizers. Direct evaluation can avoid those arbitrary choices, hence mitigating potential confounding factors and biases. Informally, direct evaluation schemes may be classified into data-driven and ground-truth based. Data-driven schemes rely on intrinsic properties computable solely on one expression matrix, such as coefficient of variation (CV) or Spearman correlation coefficient (SCC). Some comparative investigations considered the reduction of intra-condition variation, on its own, or relative to the inter-condition variation as the criterion for good normalization (*Dillies et al., 2013*; *Abrams et al., 2019*). Along the same line, the number of genes with low CV (*Glusman et al., 2013*; *Wu et al., 2019*) was used as a quantitative manifestation of such expectations. Some authors suggested the reduction in pairwise correlation among the gene profiles as an indicator normalization success (*Glusman et al., 2013*; *Wu et al., 2019*). Although such decorrelation of normalized gene expression profiles was indeed observed, complete decorrelation of all gene pairs does not necessarily coincide with optimal normalization (*Shmulevich & Zhang, 2002*).

In ground-truth based evaluation, one obtains a correctly normalized expression matrix and defines a means to quantify how close they can get to this ideal solution (Fig. 1). The most common source of ground truth is simulation which was used in many comparative studies (*Dillies et al., 2013*; *Maza et al., 2013*; *Lin et al., 2016b*; *Evans, Hardin & Stoebel, 2017*). A simulated data set allows one to perform multiple assessments on normalizers, including their impact on downstream DE detection (*Dillies et al., 2013*; *Maza et al., 2013*; *Lin et al., 2016b*; *Evans, Hardin & Stoebel, 2017*), *via* traditional measures such as false positive rate, false negative rate, false discovery rate, area under the ROC curve, etc., and their ability to recover true expression levels, *via* the mean squared error (MSE) of log-ratios, either for all genes (*Maza et al., 2013*) or for non-differential (*i.e.,* reference) genes (*Evans, Hardin & Stoebel, 2017*). Experimentally, ground truth of normalization may come from quantitative real-time polymerase chain reaction (qRT-PCR). As the only experimental technique to determine absolute expression level, qRT-PCR is often

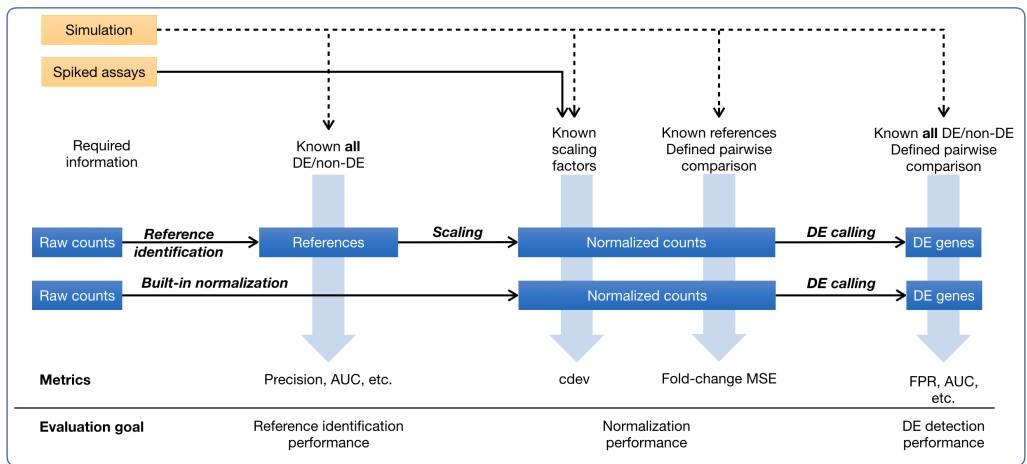

**Figure 1** **Ground-truth based evaluation methods for RNA-seq normalization, in the context of RNA-seq normalization workflows.** A normalization method may produce normalized counts in one step, or first identify the set of references which is then used for scaling the read count matrix. The assessment may happen upstream, downstream, or right on the output of normalization. The sources of ground truth are drawn as yellow boxes.

resorted to as a gold standard to judge the accuracy of gene expression estimates by RNA-seq. However, considering that qRT-PCR can only quantify one gene at a time, scaling it to thousands of genes poses significant challenges, especially in calibration across thousands of asssays (*Bustin & Nolan, 2004*). Indeed, the correlation of measurements between two qRT-PCR platforms is even lower than that between two RNA-seq platforms (*SEQC/MAQC-III Consortium et al., 2014*), making it questionable whether scaled-up qRT-PCR experiment remains as reliable. A more rigorous way to establish ground truths is the mixing of samples at pre-defined ratios to create new samples with known titrations, as done in the SEQC project (*SEQC/MAQC-III Consortium et al., 2014*). Specifically, two commercially available samples, Universal Human Reference (UHR) and Human Brain Reference (HBR), are mixed at 1:3 and 3:1 ratios to create two additional samples. These four samples together define expected titration orders and ratios which can be used to validate normalized expression levels (*Abrams et al., 2019*; *Tong et al., 2020*). Such ground truth is different from practical settings in many ways. On one hand, the RNA compositions are too different between the two samples UHR and HBR such that it is trivial for downstream applications such as DE detection, clustering, or classification. On the other hand, the variability among replicates of the same group are too small, much lesser than that usually encountered among biological replicates. Together, it is difficult to extrapolate the performance on those artificial samples into real-life setting. Yet another source of experimental ground truth are RNA-seq assays with artificial RNA spike-ins (*Jiang et al., 2011*) which must be added to biological samples at a known, constant concentration. Although there have been different opinions, both advocating for (*Lovén et al., 2012*; *Chen et al., 2016*) and cautioning against (*Qing et al., 2013*; *Risso et al., 2014b*) the use of ERCC spike-ins in normalizing RNA-seq counts, they are nevertheless the only source of

experimental ground truth by far. Spike-ins may be used very differently in processing RNA-seq data (*Risso et al., 2014b*; *Chen et al., 2016*; *Tran et al., 2020*). In regression-based normalization, a linear model is fitted on the set of spike-ins, and then used to compute the expression levels of internal genes (*Lovén et al., 2012*). In factor analysis, spike-ins are helpful as an indicator of unwanted variation which is detected and removed from the internal genes (*Risso et al., 2014a*), Lastly, spike-ins may serve as external reference genes whose counts are used to compute per-sample calibration factors (*Chen et al., 2016*; *Athanasiadou et al., 2019*; *Tran et al., 2020*) which globally scale all genes in the same sample. All three approaches would equalize the levels of spike-ins, yet the transformation is more pervasive in the first two. In the scope of this paper, we only refer to the third type of transformation, *i.e.,* global scaling, as "normalization".

The complicated landscape of performance metrics for RNA-seq normalization, as briefly surveyed above, exhibits several issues. First, comparative studies heavily depended on DE detection, an analysis downstream of normalization. In the early days of RNA-seq assays, normalization and DE detection are almost always coupled. As RNA-seq data are used in a wider range of applications, it becomes insufficient to evaluate a normalizer only in terms of DE analysis performance. Moreover, it is not practical to assess the performance of a normalizer on all possible downstream applications. Second, most performance measures in the field are data-driven. A major caveat with these metrics is the ability to optimize them directly, resulting in algorithms which will easily top the rank if judged by their own objective functions. For example, an algorithm could be devised to optimize the de-correlation of gene expression profiles (*Shmulevich & Zhang, 2002*), or to maximize the number of uniform (low CV) genes (*Glusman et al., 2013*). Apparently in such cases, these metrics are no longer an unbiased measure of normalization performance. Thirdly, almost all reputable benchmarking studies used simulation as an authoritative source of ground truth. Despite being a powerful technique, simulation depends on various modeling assumptions. Naturally, the assessment will be biased in favor of those normalizers sharing the same assumptions with the simulator.

In addressing these issues, we propose a ground-truth based measure called condition-number based deviation, or *cdev*, which measures the differences between two expression matrices. The *cdev* between a normalized expression matrix and the ground truth is then a succinct indicator of normalization performance. *cdev* was first utilized by *Tran et al. (2020)* to validate a new normalization method on real data sets. In this paper, we elaborated on its mathematical properties, showing that it can be computed for any two expression matrices of the same shape, and is symmetric as long as the normalizing operation only involves scaling the samples. We demonstrated that *cdev* is indeed concordant with normalization quality when measured by both upstream and downstream criteria. In companion with *cdev*, we compiled a collection of spiked RNA-seq assays which can serve as experimental ground truth for normalization. This toolset will assist in the validation and comparison of RNA-seq normalizers, independent from downstream analyses and unbiased toward any particular generative model of RNA-seq read counts.

**Table 1   Summary of simulations used in the study.**

| Inspection intent | Number of samples in condition 1 | Number of samples in condition 2 | Number of genes | Fraction of DE genes | Fold-change between two conditions | Simulation type |
|---|---|---|---|---|---|---|
| *cdev* as a function of normalization quality | 20 | 20 | {2000, 10000, 20000, 50000} | 0.5 | 4 | Homogeneous fold-change |
| | 20 | 20 | 20000 | 0.5 | {1.5, 2, 3, 4, 5, 6, 7, 8} | Homogeneous fold-change |
| | 20 | 20 | 10000 | {0.1, 0.2, 0.3, 0.4, 0.5, 0.6, 0.7, 0.8, 0.9} | 4 | Homogeneous fold-change |
| | 20 | 20 | {2000, 10000, 20000, 50000} | 0.5 | Continuous | Heterogeneous fold-change |
| | 20 | 20 | 10000 | {0.1, 0.2, 0.3, 0.4, 0.5, 0.6, 0.7, 0.8, 0.9} | Continuous | Heterogeneous fold-change |
| Concordance of *cdev* and fold-change MSE | 15 | 15 | 4000 | {0.1, 0.2, 0.3, 0.4, 0.5, 0.6, 0.7, 0.8, 0.9} | Continuous | Heterogeneous fold-change |
| Use of *cdev* to compare normalization methods | 20 | 20 | 10000 | {0.1, 0.2, 0.3, 0.4, 0.5, 0.6, 0.7, 0.8, 0.9} | Continuous | Heterogeneous fold-change |

## MATERIALS & METHODS

### Simulations

Simulation of RNA-seq counts was performed in the same manner as done by *Law et al. (2014)* and later by *Evans, Hardin & Stoebel (2017)*. Specifically, baseline relative abundance was generated for $n$ genes in $m$ samples of two conditions, according to an empirical distribution obtained from real data (*Law et al., 2014*). In $m/2$ samples of condition 2, a $k$ percentage of these values were then multiplied by the fold-change $f$, creating $k \times n$ DE genes. From these baselines, counts were generated by sampling from the corresponding negative binomial distribution. There are two types of simulations used in this study. In the homogeneous fold-change scenario, a single fold-change was applied for all DE genes. For a more realistic scenario, *i.e.,* heterogeneous fold-change, random values of fold-change were sampled from the uniform distribution $Unif(1, 10)$ for up-regulated genes, or $Unif(-10, -1)$ for down-regulated genes, such that 50% of the DE genes are up-regulated, and 50% are down-regulated. This choice is arbitrary, since the distribution of fold-change in real data sets comes in all shapes and sizes, and even shifted by normalization.

A summary of different simulation scenarios were given in Table 1.

### Compilation of RNA-seq data

RNA-seq data were collected from the Gene Expression Omnibus (GEO) database based on the following criteria: having ERCC spike-ins added at a constant concentration across a sufficiently large number of samples. The list of studies where these assays were originally performed is available in Table 2.

**Table 2  Original studies where RNA-seq assays were performed with ERCC spike-ins.**

| Name | Species | Source | GEO accession | Genome assembly | R data package | Data sets |
|---|---|---|---|---|---|---|
| Rat Bodymap (*Yu et al., 2014a*; *Yu et al., 2014b*) | *Rattus norvegicus* | Tissues | GSE53960 | Ensembl Rnor 6.0 | data.rnaseq. RnorBodymap | rat1, rat2 |
| SEQC Toxicogenomics (*Wang et al., 2014*; *Bushel, Paules & Auerbach, 2018*; *Gong et al., 2014*) | *Rattus norvegicus* | Livers | GSE55347 | Ensembl Rnor 6.0 | data.rnaseq. Rnor | rnor1, rnor2 |
| Sarcoma (*Lesluyes et al., 2016*; *Lesluyes et al., 2019*) | *Homo sapiens* | Sarcoma samples | GSE71119 | GRCh38.p10 | data.rnaseq. sarcoma | sarcoma |
| Yellow-fever-virus infected cells | *Homo sapiens* | Hepatocytes and Kupffer cells | GSE99081 | GRCh38.p10 | data.rnaseq. YFV | yfv |
| Lymphoma (*Zhao et al., 2019*) | *Homo sapiens* | B-cell lymphomas | GSE116129 | GRCh38.p10 | data.rnaseq. lymphoma | lymphoma |
| Frog development (*Owens et al., 2016*) | *Xenopus tropicalis* | Embryos | GSE65785 | Ensembl Xtro 9.1 | data.rnaseq. XtroDev | xtro1m, xtro2 |
| Fly (*Lin et al., 2016a*; *Lin et al., 2016b*) | *Drosophila melanogaster* | Whole animals | GSE60314 | Ensembl BDGP6.28 | data.rnaseq. Dmel | dmelAC, dmelAV |
| Mouse brain during pregnancy and postpartum (*Ray et al., 2015*) | *Mus musculus* | Mouse brain regions | GSE70732 | GRCm38.p6 | data.rnaseq. MmusPreg | mpreg1a, mpreg1b, mpreg2a, mpreg2b |

All RNA-seq assays from said studies were downloaded from the Sequence Read Archive (SRA) in FASTQ format and re-processed by a single workflow. In this workflow, the raw reads were aligned to the corresponding transcriptome using STAR v2.7 (*Dobin et al., 2013*), and quantified using RSEM v1.3 (*Li & Dewey, 2011*). Prior to alignment and quantification, each species-specific transcriptome was extended by a set of 92 ERCC spike-in sequences (*Jiang et al., 2011*) to create the software-specific indices. Metadata were extracted and compiled manually from all available information in GEO, SRA, the sequence identifiers in the FASTQ files, and original publication when necessary. The biological samples were then separated into individual data sets, such that every sample in the same set was added with the same amount of ERCC spike-ins being processed in the same experimental procedure (Table 3). The data sets from each project were then wrapped in R data packages (Table 2) to facilitate further analysis. All data packages are publicly available on GitHub.

## Ground-truth normalization

An expression profiling experiment of $n$ genes on $m$ samples typically results in an $n \times m$ matrix $\mathbf{Y}$, the element $y_{ij}$ of which captures the measurement of gene $i$ in sample $j$. In RNA-seq, $\mathbf{Y}$ is the read count matrix output by the quantification process. To enable between-sample comparison, the read counts in $\mathbf{Y}$ requires proper normalization. The normalized expression matrix $\mathbf{X}$ is obtained by scaling $\mathbf{Y}$ against a set of $m$ factors $v = \{v_1, v_2, \ldots, v_m\}$ such that every gene in sample $j$ is scaled by the same factor $v_j$. Let

**Table 3  Compilation of re-processed RNA-seq data sets for use as ground-truths in normalization.**

| Data set label | Experimental design | Number of genes | Number of samples | Number of good references | Spike-ins |
|---|---|---|---|---|---|
| dmelAC | 2 environments × 16 strains × 2 sexes | 17860 | 367 | 7 | ERCC 78A, centrifuge |
| dmelAV | 2 environments × 16 strains × 2 sexes | 17860 | 165 | 15 | ERCC 78A, vacuum |
| mpreg1a | 3 brain regions × 3 post-partum stages | 55493 | 27 | 14 | ERCC, 1:10 dilution |
| mpreg1b | 1 brain region × 3 post-partum stages | 55493 | 9 | 15 | ERCC, 1:100 dilution |
| mpreg2a | 3 brain regions × 3 stages | 55493 | 27 | 16 | ERCC, 1:10 dilution |
| mpreg2b | 1 brain region × 3 stages | 55493 | 9 | 17 | ERCC, 1:100 dilution |
| rbm1 | 5 organs × 4 stages × 2 sexes | 32975 | 159 | 28 | ERCC Mix 1 |
| rbm2 | 6 organs × 4 stages × 2 sexes | 32975 | 159 | 27 | ERCC Mix 2 |
| rnor1 | 26 chemicals | 32975 | 64 | 37 | ERCC Mix 1 |
| rnor2 | 16 chemicals | 32975 | 52 | 37 | ERCC Mix 2 |
| sarcoma | 9 diseases × 2 metastatic states | 58380 | 125 | 36 | ERCC Mix, Life Technologies |
| lymphoma | 4 cell lines × 3 treatments | 58380 | 34 | 11 | ERCC Mix 1, 1 μL to 5000 ng RNA |
| yfv | 2 cell models × 3 infections | 58380 | 24 | 21 | ERCC Mix 1 |
| xtro1m | Time series, replicated in 2 clutches | 21550 | 145 | 16 | ERCC Mix 1, polyA enrichment |
| xtro2 | Time series | 21550 | 48 | 28 | ERCC Mix 1, RiboZero enrichment |

$R$ be the set of reference genes whose expression levels are stable across all samples. If this set is identified, the scaling factor $v_j$ can be computed for each sample as follows

$$v_j = \frac{\sum_{i \in R} y_{ij}}{\left(\prod_{k=1}^{m} \sum_{i \in R} y_{ik}\right)^{1/m}}.$$

Note that any multiplication of $v$ by a scalar results in a valid normalization. To obtain expression levels at a comparable magnitude with the raw counts, we chose to scale these factors by their geometric mean. This step facilitates the visual comparison of expression matrices but does not critically affect the outcome of differential expression analysis (Fig. S8).

In simulations, the reference set $R$ is composed of all non-DE genes, *i.e.,* those with fold-change set to one. In real data sets, $R$ contains the ERCC spike-ins detected at sufficiently high number of read counts, that is, at least $t$ reads in every sample. The threshold $t$ ranges from 10 to 100 depending on the data sets, in order to eliminate unreliable measurements while retaining enough references to compute normalization factors. Manual examination of each data set was documented in the R notebook published as part of the source code.

## Condition-number based deviation (*cdev*)

**Definition**. Assuming that a ground-truth expression matrix, denoted **A**, is known, the quality of the normalized expression matrix **X** can then be measured by the condition-number based deviation (*cdev*) between **X** and **A**, denoted *cdev*(**X**, **A**). To compute *cdev*(**X**, **A**), we first define the transformation from **X** to **A**. Let **B** be the matrix that transforms **X** to **A**, *i.e.,* **XB** = **A**. As we are interested in recovering the pattern of differential expression rather than the absolute concentration of each RNA species, any multiplication

of the ground truth $\mathbf{A}$ by a scalar is considered a valid normalization. Equivalently, $\mathbf{X}$ is considered valid if $\mathbf{B} = \alpha \mathbf{I}$, where $\alpha$ is any scalar. Hence the quality of $\mathbf{X}$ can be measured by how much $\mathbf{B}$ deviates from the identity form. Such deviation is quantified by the condition number of $\mathbf{B}$, defined as $\kappa(\mathbf{B}) = \|\mathbf{B}\| \cdot \|\mathbf{B}^{-1}\|$. Choosing $\|\bullet\|$ to be $\ell_2$ norm, the condition number can be calculated using the ratio between the largest and smallest singular values, that is, $\kappa(\mathbf{B}) = \frac{\sigma_{max}(\mathbf{B})}{\sigma_{min}(\mathbf{B})}$. To compute $\mathbf{B}$, one may recognize that $\mathbf{B} = (\mathbf{X}^T\mathbf{X})^{-1}\mathbf{X}^T\mathbf{A}$, in which $(\mathbf{X}^T\mathbf{X})^{-1}\mathbf{X}^T$ is the Moore–Penrose pseudoinverse of $\mathbf{X}$, denoted as $\mathbf{X}^\dagger$. Hence the definition of *cdev* can be written as follows

$$cdev(\mathbf{X}, \mathbf{A}) \equiv \kappa(\mathbf{X}^\dagger\mathbf{A}) = \frac{\sigma_{max}(\mathbf{X}^\dagger\mathbf{A})}{\sigma_{min}(\mathbf{X}^\dagger\mathbf{A})} \quad .$$

The pseudoinverse $\mathbf{X}^\dagger$ can be computed using the singular value decomposition $\mathbf{X} = \mathbf{U}\Sigma\mathbf{V}^T$, that is, $\mathbf{X}^\dagger = \mathbf{V}\Sigma^{-1}\mathbf{U}$ (*Alter & Golub, 2004*; *Alter et al., 2004*).

**Existence and uniqueness**. The existence of $cdev(\mathbf{X}, \mathbf{A})$ depends on that of the pseudoinverse $\mathbf{X}^\dagger$ and the singular value decomposition of $\mathbf{X}^\dagger\mathbf{A}$, both of which are guaranteed to exist. Hence $cdev(\mathbf{X}, \mathbf{A})$ always exists for any expression matrices $\mathbf{X}$ and $\mathbf{A}$. As the singular values of $\mathbf{X}^\dagger\mathbf{A}$ are unique, so is $cdev(\mathbf{X}, \mathbf{A})$.

**Symmetry**. One should note that, *cdev* is not symmetric, *i.e.*, $cdev(\mathbf{X}, \mathbf{A}) \neq cdev(\mathbf{A}, \mathbf{X})$ for generally given matrices $\mathbf{X}$ and $\mathbf{A}$. However, when the transformation from $\mathbf{X}$ to $\mathbf{A}$ involves only column-scaling, we have $cdev(\mathbf{X}, \mathbf{A}) = cdev(\mathbf{A}, \mathbf{X})$. To see why, we first notice that the transformation matrix $\mathbf{B}$, by which $\mathbf{XB} = \mathbf{A}$, is now diagonal,

$$\mathbf{B} = \begin{bmatrix} b_{11} & 0 & \dots & 0 \\ 0 & b_{22} & \dots & 0 \\ \vdots & \vdots & \ddots & \vdots \\ 0 & 0 & \dots & b_{mm} \end{bmatrix}.$$

The transformation from $\mathbf{A}$ to $\mathbf{X}$ is then $\mathbf{C} = \mathbf{B}^{-1}$. By definition of *cdev*,

$$cdev(\mathbf{A}, \mathbf{X}) = \kappa(\mathbf{C}) = \kappa(\mathbf{B}^{-1}) = \frac{1/\sigma_{min}(\mathbf{B})}{1/\sigma_{max}(\mathbf{B})} = \kappa(\mathbf{B}) = cdev(\mathbf{X}, \mathbf{A}).$$

Consequently, when used to measure the changes in expression matrices made by global scaling, $cdev(\mathbf{X}, \mathbf{A})$ is a symmetric measure.

**Interpretation**. *cdev* ranges from 1 to infinity, with a smaller *cdev* indicating more similar expression patterns. The definition of *cdev* is closely related to that of a valid normalization. Specifically, any multiplication $\mathbf{X}'$ of the matrix $\mathbf{X}$ by a scalar results in $cdev(\mathbf{X}, \mathbf{X}') = 1$.

Because any non-empty subset of references result in a valid normalization, the result of scaling $\mathbf{Y}$ by the full set of references $R$, denoted $\mathbf{X}_R$, should be similar to that of scaling by the subset $S \subset R$, denoted $\mathbf{X}_S$. Hence, $cdev(\mathbf{X}_S, \mathbf{X}_R)$ should be close to 1. This implication is helpful in validating the behavior of *cdev* on real data sets (See the Results section).

**Implementation**. Although $cdev(\mathbf{X}, \mathbf{A})$ can be computed for any $\mathbf{X}$ and $\mathbf{A}$ of the same shape using the definition above, it is much more trivial and efficient to compute in practice, when $\mathbf{X}$ is transformed to (or from) $\mathbf{A}$ by global scaling. In particular, the singular values of diagonal matrix $\mathbf{B}$ are now its elements. These elements also define the normalization vector $\mathbf{v}$ which can be re-constructed by dividing an arbitrary row $\mathbf{X}[i, ]$ of $\mathbf{X}$ by the corresponding row $\mathbf{A}[i, ]$ of $\mathbf{A}$, as long as the feature $i$ is not zero in any sample.

## Normalization methods

As an illustration for the use of *cdev*, we compared six common normalization methods, including Total Count (TC), Upper Quartile (UQ) (*Bullard et al., 2010*), trimmed mean of M-values (TMM) (*Anders & Huber, 2010*), DESeq (*Love, Huber & Anders, 2014*), DEGES/TMM (Differentially Expressed Gene Elimination Strategy) (*Kadota, Nishiyama & Shimizu, 2012*), and PoissonSeq (*Li et al., 2012*). These methods were chosen due to the simplicity of computation and/or availability of R implementation which allows quick integration to the current analysis.

Each of these methods produce an $m$-vector $\mathbf{v}$, the element $v_j$ of which is used to scale sample $j$. While some methods produce close-to-1 scaling factors, some others require further normalization of these factors, for example, by the geometric mean of the elements $v_j$, to keep the magnitude comparable. As the magnitude of $\mathbf{v}$ does not affect the validity of its resulting normalization, a simplified description of $\mathbf{v}$ resulted by each method is provided below to highlight the differences between them.

In TC, the scaling factor $v_j$ is proportional to the total count of all genes measured in sample $j$, *i.e.*,

$$v_j^{\mathrm{TC}} \propto \sum_{i=1}^{n} y_{ij}.$$

In UQ, the factor is computed by the upper quartile instead of the summation over gene counts,

$$v_j^{UQ} \propto \text{Upper-Quartile}_{i \in G^*} y_{ij},$$

in which $G^*$ is the set of genes with reads in at least one sample.

In TMM, the scaling factor is computed from the ratio of a gene count in sample $j$ and the reference sample, arbitrarily selected as the first one,

$$v_j^{\mathrm{TMM}} \propto \frac{1}{|G^{\mathrm{TMM}}|} \sum_{i \in G^{\mathrm{TMM}}} \frac{y_{ij}}{y_{i1}},$$

in which $G^{\mathrm{TMM}}$ denotes the set of genes retained by TMM criteria.

In DESeq, scaling factors are computed from the medians of the ratio between a count in sample $j$ and that in the pseudo-reference sample, which is computed by taking the geometric mean of each gene over the samples,

$$v_j^{\mathrm{DESeq}} \propto \mathrm{median}_i \left[ \frac{y_{ij}}{\left( \prod_{k=1}^{m} y_{ik} \right)^{1/m}} \right].$$

DEGES is an iterative strategy in which the count matrix is scaled by a normalization method (such as TMM, DESeq, etc.), DE genes are identified by a DE caller (such as edgeR, DESeq, etc.), and the normalization is repeated with DE genes now removed (*Kadota, Nishiyama & Shimizu, 2012*; *Sun et al., 2013*). There are multiple choices for the normalizer and the DE caller, as well as the number of iterations. In this work, we chose TMM as the first-step normalizer, and edgeR as the second-step DE caller. With one iteration, the workflow is TMM-edgeR-TMM, which we denoted as DEGES/TMM. The scaling factor at iteration $t$ is computed using the same method as that of TMM, while restricting the set of genes to that of $G^t$,

$$v_j^{\text{DEGES/TMM}}(t) \propto \frac{1}{|G^t|} \sum_{i \in G^t} \frac{y_{ij}}{y_{i1}},$$

in which $G^t$ is the set of genes identified as non-DE at iteration $t$.

PoissonSeq is also an iterative protocol which starts with fitting a Poisson log linear model under the null hypothesis that no gene is DE, then removes those with low ranking for goodness-of-fit, and fits again under the new hypothesis until convergence. The scaling factor is computed in the similar manner to that of TC,

$$v_j^{\text{PoissonSeq}} \propto \frac{\sum_{i \in G^t} y_{ij}}{\sum_{k=1}^{m} \sum_{i \in G^t} y_{ik}},$$

in which $G^t$ is the set of genes remained at iteration $t$.

## Differential expression analysis

Differential expression (DE) analysis was used as a downstream indicator of normalization quality. Among a great number of methods developed for calling DE genes, we chose *limma* in consideration of its speed and robust performance relative to other methods (*Soneson & Delorenzi, 2013*). The method was implemented in the R package `limma` (*Law et al., 2014*; *Ritchie et al., 2015*).

Prior to DE calling, all the raw counts were increased by a small count (averaging 1), log-transformed, and normalized, using the function `edgeR::cpm()` from the `edgeR` package (*Robinson, McCarthy & Smyth, 2010*). The resulted log-cpm matrix was then passed to `limma` functions, following the *limma-trend* protocol (*Law et al., 2014*).

To verify that the magnitude of scaling factors does not affect the outcome of DE analysis regardless of the specific choice of DE caller, we performed DE gene detection using three methods *limma-trend*, *DESeq* (*Love, Huber & Anders, 2014*) and *edgeR* (*Robinson, McCarthy & Smyth, 2010*), as implemented in the R packages `limma`, `DESeq2`, and `edgeR`, respectively.

All the analyses were done in R computing environment (*R Core Team, 2021*).

## RESULTS

### Real data sets as experimental ground-truth

We presented here a diverse compilation of RNA-seq data sets with ERCC spike-ins, covering five different species: *Homo sapiens*, *Rattus norvegicus*, *Mus musculus*, *Xenopus*

*tropicalis*, and *Drosophila melanogaster*. This collection is by far the most diverse set of publicly available RNA-seq assays with ERCC spike-ins.

Some earlier observations were reproduced on these data sets. First, the read counts of ERCC spike-ins are highly correlated with one another, especially those at high and moderate levels. This phenomenon was consistently observed in all data sets (Fig. S2), corroborating what was predicted theoretically and observed empirically on a mouse data set (*Tran et al., 2020*). Second, differences in library preparation protocol led to uneven efficiency among the spike-in sequences. Such protocol-dependent biases were observed earlier by *Qing et al. (2013)* (on data sets not available publicly) and by *Owens et al. (2016)* (on frog development time series re-processed into *xtro1m* and *xtro2* in this work). Earlier studies such as *Qing et al. (2013)* and *Risso et al. (2014a)* also noted the fluctuation of reads mapped to ERCC spike-ins across all samples. Although we did find the same behavior, it should be noted that such fluctuation is an expected behavior of un-normalized read counts, and does not invalidate the use of these sequences as experimental references (Fig. S1).

Within each data set, the samples were spiked with the same concentration of ERCC spike-in RNAs and prepared with the same experimental protocol. Consequently, spike-ins are the true references in each data set. After normalizing by these references (see Methods section for the exact formula), the levels of ERCC spike-ins become well equalized (last columns in Fig. S3).

### *cdev* is indicative of both upstream and downstream performance

In order for *cdev* to be a useful metric, it must be indicative of normalization quality which may be controlled by the quality of the reference set. It was shown that including differential genes in this set results in an invalid normalization (*Tran et al., 2020*). Using simulated data, it was clear that *cdev* is linearly correlated with the reference set quality, that is, inversely proportional to the fraction of differential genes included in the scaling set (Fig. 2, left column). This is true in both simulation scenarios, whether the fold-change is fixed for all DE genes (homogeneous fold-change), or varied over a continuous range (heterogeneous fold-change).

In line with conventional assessment, we expect *cdev* to be predictive of DE detection performance, for the improvement of normalization quality should lead to that of DE detection.

On the same simulated data as mentioned above, *cdev* is highly indicative of DE calling success which is summarized by the area under the receiver-operating characteristic (ROC) curve (AUC) (Fig. 2). When fold change is fixed for all DE genes, AUC as a function of *cdev* has a reverse sigmoid shape. The performance of DE calling drops significantly when *cdev* reaches a high enough threshold (Figs. 2A–2F). When fold change of DE genes spans a continuous range, this relation gets closer to a linear form (Figs. 2G–2J).

*cdev* is agnostic to the magnitude of scaling factors, as are DE gene calling procedures. This behavior is demonstrated empirically on simulated data sets. DE performance is almost identical on expression matrices scaled by the same-direction vectors spanning

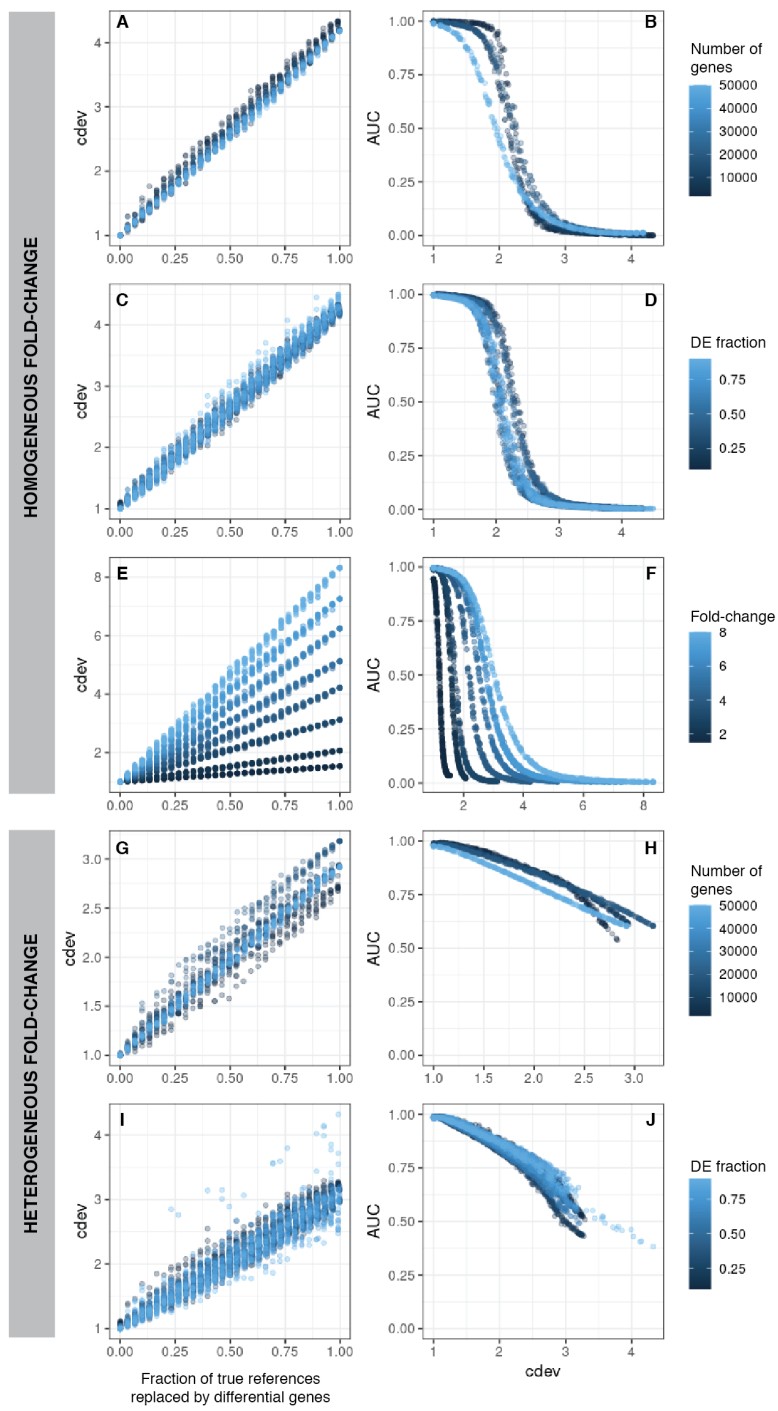

**Figure 2** ***cdev* as a function of normalization quality on simulation data.** A, C, E, G, I: *cdev* as the function of reference set quality, measured in the fraction of true differential genes. B, D, F, H, J: *cdev* as a predictor of DE detection performance. (A-F) The relationship on homogeneous fold-change simulation; (G-J) the relationship on heterogeneous fold-change simulation.

several orders of magnitude. Not only *limma*, but also count-based methods such as *DESeq* and *edgeR*, were not affected by the magnitude of scaling factors (Fig. S8).

Simplistic simulation enables us to tear apart the effects of different data characteristics on *cdev*. By varying one parameter at a time, we found that the range of *cdev* is not affected by the number of genes, nor the fraction of DE genes, but determined by the level of difference among the samples, represented by the fold change between two simulated conditions (Figs. 2A, 2C, 2E). Fold-change also dictates how tolerant DE detection can be toward bad normalization. Specifically, when fold change is as high as 8, a normalized expression matrix with *cdev* of 3 can still result in an AUC larger than 0.5. At smaller fold change such as 2, this level of deviation from the ground truth is sufficient to drop the AUC of DE detection to zero. This behavior makes sense, for it is easier to call DE when the fold change is large.

Since the behavior of *cdev* on real data sets could not be validated with the same metrics as that on simulation data, we examined if it behaves as theoretically predicted. Specifically, $cdev(\mathbf{A}', \mathbf{A})$ should be close to 1 with $\mathbf{A}$ being normalized by the true reference set $R$, and $\mathbf{A}'$ by any non-empty subset $S$ of $R$. Meanwhile, $cdev(\mathbf{A}_{rand}, \mathbf{A})$ where $\mathbf{A}_{rand}$ is resulted from a random set of genes should be larger. These expectations were observed in all the real data sets (Fig. 3). In highly similar samples, such as those from brain regions of mice with identical genetic background (data sets *mpreg1a*, *mpreg1b*, *mpreg2a*, *mpreg2b*), $cdev(\mathbf{A}_{rand}, \mathbf{A})$ is larger than $cdev(\mathbf{A}', \mathbf{A})$, but never exceeds 10. In highly different samples, such as those from 16 fly strains (*dmelAC*, *dmelAV*), random normalization results in very large deviation from the ground truth, with $cdev(\mathbf{A}_{rand}, \mathbf{A})$ mostly around 100. Data sets with average heterogeneity, such as those of rat tissues (*rbm1*, *rbm2*) or sarcoma samples (*sarcoma*), exhibit $cdev(\mathbf{A}_{rand}, \mathbf{A})$ in the middle range.

These observations illustrate the value range of *cdev* in practice, at the same time implies the importance of normalization with respect to the heterogeneity of transcriptomic data. Specifically, when the expression profiles are highly similar, wrong normalization is not as detrimental as it would be on heterogeneous ones. Based on the behavior of *cdev* on real data sets (Fig. 3), we may recommend acceptable values of *cdev* below which DE performance is better than random. Specifically, on extremely diverse expression profiles (*dmelAC*, *dmelAV*), *cdev* up to 10 may be acceptable for DE analysis. On moderately diverse expression profiles (*rbm1*, *rbm2*, *rnor1*, *rnor2*, *sarcoma*), *cdev* below 5 may be necessary to obtain good enough DE calling. On closely related expression profiles (*mpreg1b*, *mpreg2b*, *yfv*), *cdev* should be as low as 2 in order to provide a good enough DE results. As *cdev* is a continuous measure of normalization quality, these values should not be treated as hard threshold, but rather a guideline to understand the normalization outcome. The intent of *cdev* is not to predict downstream analysis performance, but rather to enable the measurement of normalization quality independent from downstream analyses, as would be argued in the Discussion section.

### *cdev* as a robust alternative to fold-change MSE

Among a few ground-truth based measures of normalization quality, fold-change MSE is an intuitive metric. As the name implies, a fold-change MSE closer to zero indicates a

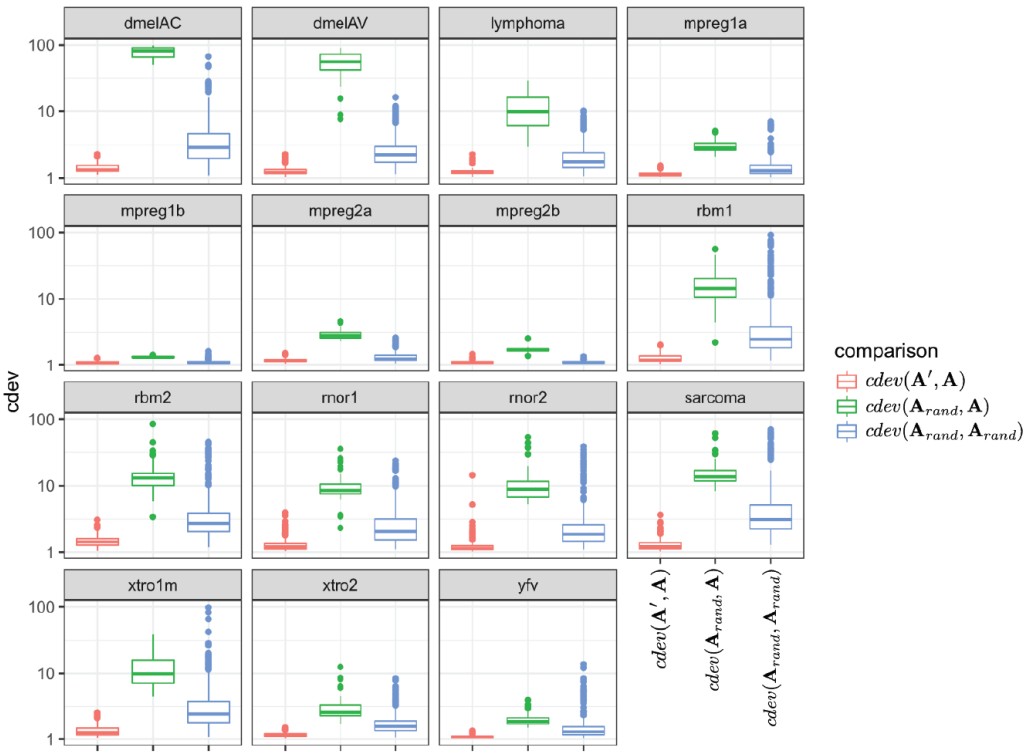

**Figure 3   The behavior of *cdev* on various real data sets.** In the labels, **A** denotes the ground truth which is obtained by normalizing against $R$, the full set of reliably detected ERCC spike-ins in each data set, **A′** the one resulted from normalizing against $S \subset R$, and **A**$_{rand}$ from normalizing against a random set of genes.

better normalization. In *Maza et al. (2013)*, fold-change MSE was defined for all genes, while in *Evans, Hardin & Stoebel (2017)*, it was narrowed down to the true reference (*i.e.,* non-DE) genes. The latter version requires less information from the ground truth and works equally well as the former, hence chosen to compare with *cdev*. When defined for true references, the expected fold-changes are all one.

In order to compare the two metrics on simulation data, we normalized an expression matrix by random sets of genes, computed both *cdev* and fold-change MSE with respect to the corresponding ground truth. To ensure these randomly normalized matrices cover a wide enough range of quality, we varied the fraction of differential genes allowed in the set of scaling genes. On this simulated data, the two metrics are highly correlated (Fig. 4).

On real data, the relationship becomes more noisy (Fig. 5). On a closer look, the "noise" is actually caused by multiple slopes of the linear relation between *cdev* and fold-change MSE. This complicated behavior happens due to the ambiguity of fold-change MSE by which many improper normalizations are able to score zero fold-change error (Fig. 6E). The problem is exaggerated on multi-factorial experiments, such as that of the lymphoma data set. In this experiment, different cell lines originating from the parental cells were treated with THZ1 (*Zhao et al., 2019*), resulting in samples of varied cell lines and THZ1 concentrations. When the data set is composed of samples from a single cell line, such as

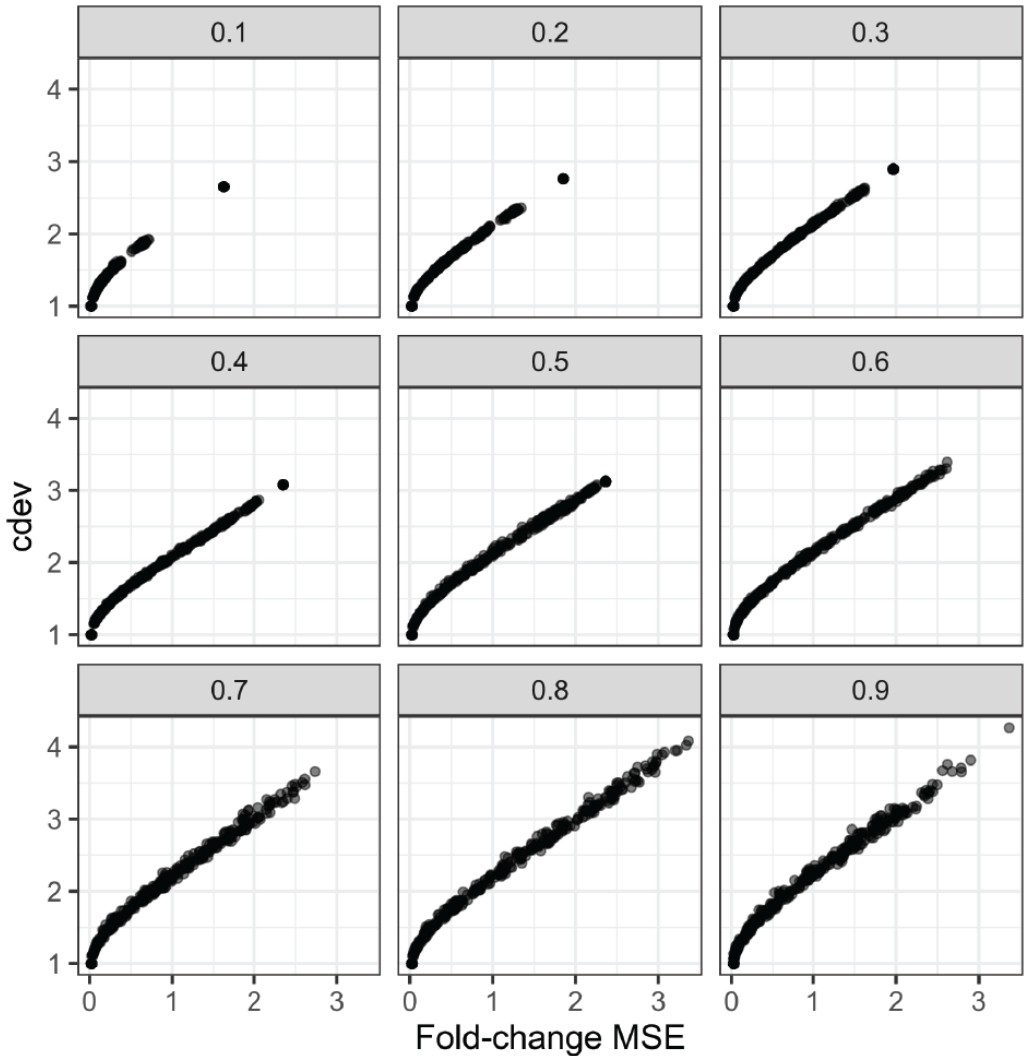

**Figure 4** Relationship between *cdev* and fold-change MSE in simulation, at varying fraction of DE genes.

I10 or L20, the relationship between *cdev* and fold-change MSE is cleanly linear. When the samples include those from the two cell lines, the two linear functions with different slopes aggregate and make the correlation seem noisy (Fig. 5). Discordance between the two metrics is caused by large *cdev* and small MSE. One such example was visualized in Fig. 6. In this case, the random set of genes could not equalize the true references, nor themselves, resulting in a poorly normalized expression matrix. This faulty result was able to score a low fold-change MSE, but not a low *cdev*. Projections on the first two principal components of the ERCC spike-ins showed that random references could not remove the variation between two organs (heart and kidney), while the true ones properly removed such variation. Although there may seem to be a few instances with "high" fold-change MSE and "low" *cdev*, they are in fact part of a *cdev*-MSE function with a lower slope, thus

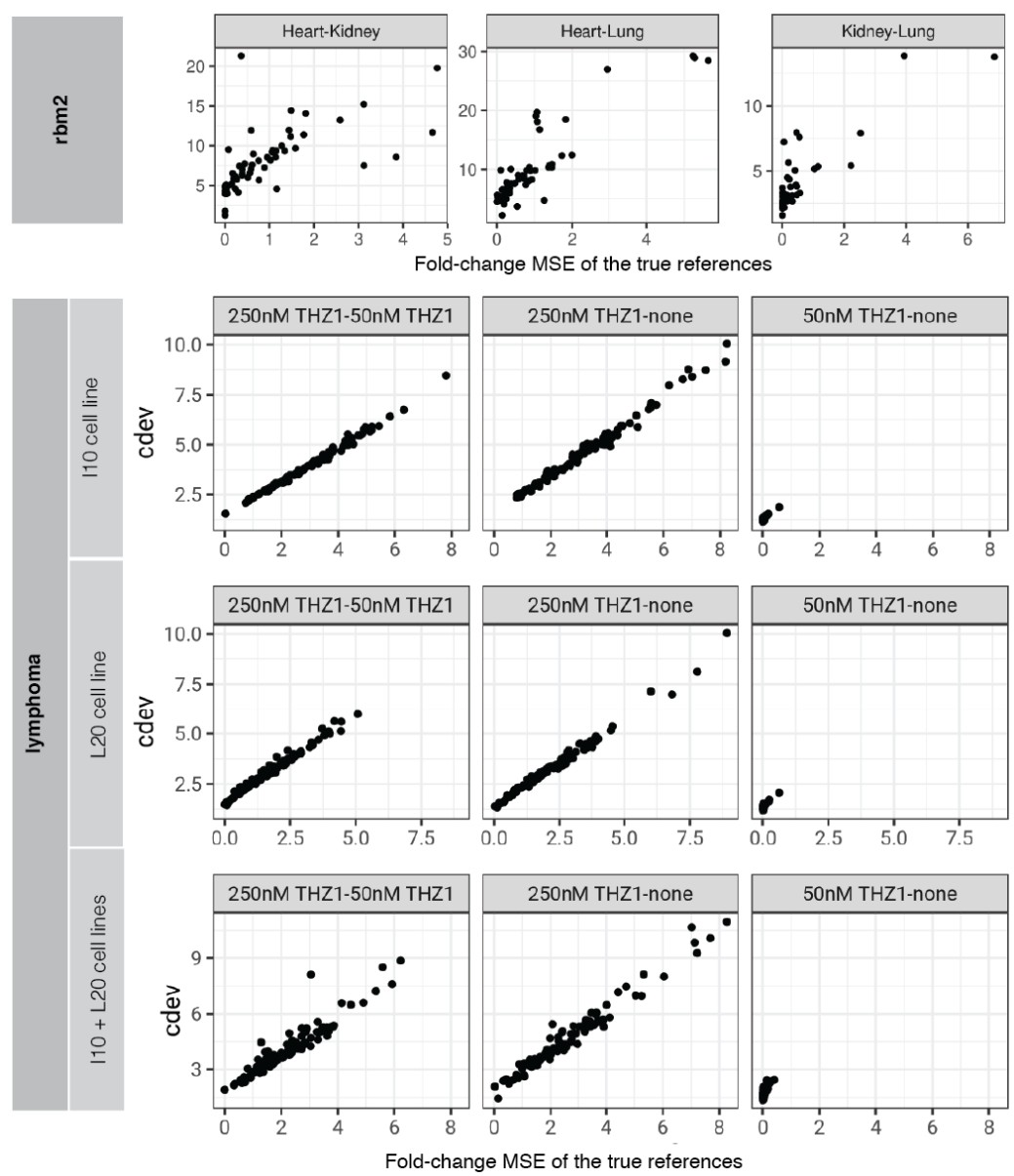

**Figure 5  Relationship between *cdev* and fold-change MSE in practice.** Upper panel: *cdev* as a function of fold-change MSE in representative pairs of organs in Rat Bodymap data set, Mix 2. Lower panels: *cdev* as a function of fold-change MSE in pairs of experimental conditions in the lymphoma data set, on samples of I10 cell line, L20 cell line, and the mix of two cell lines. The mixing of samples from two cell lines (I10 + L20) emulates the increase in heterogeneity of transcriptomic profiles.

do not contribute to the discordance between the two metrics. Inspection of such cases confirm that these random normalizations are clearly worse than the true one (Fig. S7).

## Application of *cdev* to compare common normalization methods

*cdev* is specifically devised to measure performance by global scaling normalization. To illustrate its use, we compare six common normalizers. Four of these are one-step

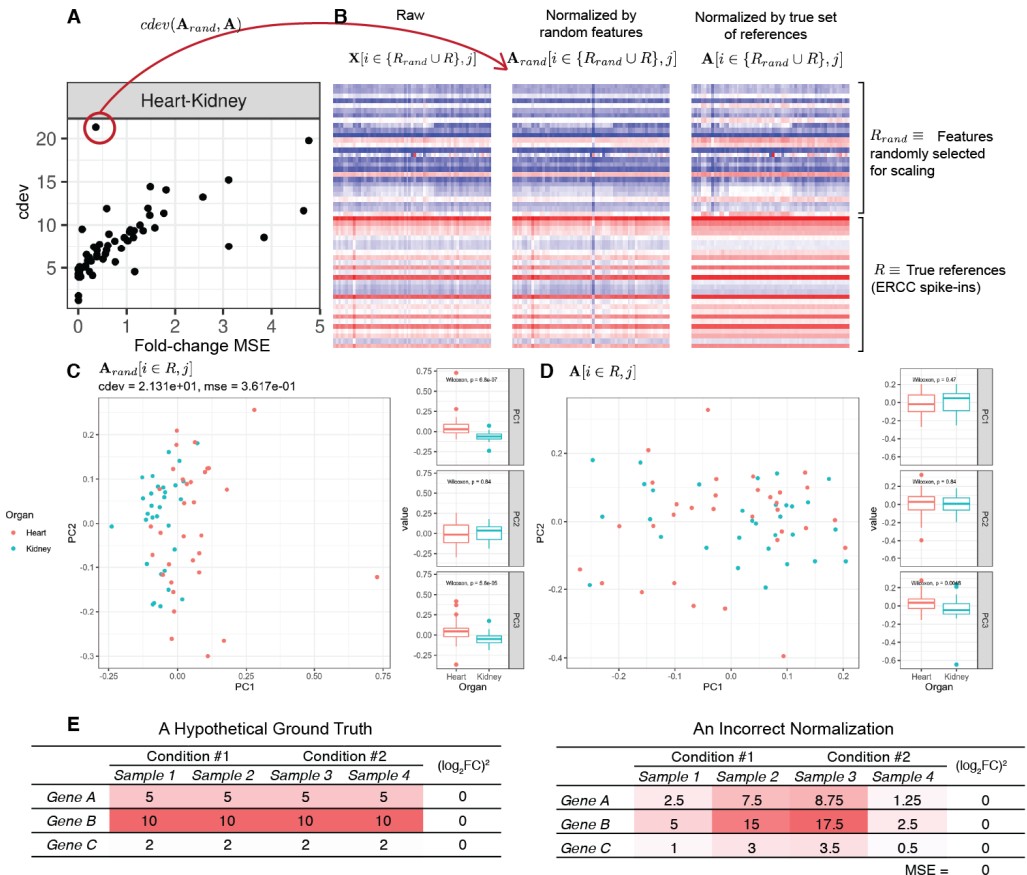

**Figure 6  A close-up examination of discordance between *cdev* and fold-change MSE on real data sets.** In labels of the plots, **A** denotes the ground truth obtained by normalizing against the true references $R$, $\mathbf{A}_{rand}$ the result of normalizing against a random set of genes $R_{rand}$ which has the same number of genes as $R$, and $\mathbf{X}$ the un-normalized read count. In all cases, the column index $j$ implies the selection of samples from heart and kidney. (A) The relation between *cdev* and fold-change MSE on a subset of the *rbm2* (Rat Bodymap, ERCC Mix 2) data set, with an outlier scoring a small fold-change MSE and a very large *cdev*. (B) This outlier was the result of normalization by a random set of genes which could not equalize the true references nor themselves. (C) Projection of $\mathbf{A}_{rand}[i \in R, j]$ on the first two principal components (PC) reveals that organ difference is still the dominant variation on the spike-ins after this random scaling, as opposed to (D) the correct normalization in which the spike-ins are no longer the source of variation between the two organs. (E) A hypothetical normalization explaining why fold-change MSE can be ambiguous. In this example, three reference genes are all normalized incorrectly, yet resulting in no fold change between two conditions, leading to zero fold-change MSE.

methods in which the scaling factors are computed using a closed form formula. The other two (DEGES/TMM and PoissonSeq) are iterative procedures in which the set of non-differentially expressed features are refined until convergence. In all the following evaluation, the ground truth is computed by scaling against the set of known references. For simulated data, this set includes the genes with fold-change set to 1 by the simulator, such that the number of DE (and non-DE) genes satisfies the required percentage of differential expression. For real data, it includes selected ERCC spike-ins which have sufficiently high number of reads. In both simulation and real data, the true references were able to equalize

themselves across all samples (Figs. 7 and 8). All normalizers perform worse when the input data becomes more challenging, *i.e.,* contains a larger fraction of DE genes. At the highest DE fraction simulated, none of the normalizers could stabilize the expression levels of non-DE (*i.e.,* reference) genes (Fig. 7D). As the quality of normalization decreases, so is the performance of DE detection. The false positive rate (FPR) and AUC of DE calling similarly decreases with increasing DE percentage, demonstrating a linear relationship with *cdev* (Fig. 7E). Although false negative rate (FNR), *i.e.,* miss rate, was used in previous studies (*Maza et al., 2013*), we found it not informative for DE detection performance (Fig. S5). Apparently, one may achieve zero FNR by calling all genes differential. Meanwhile, incorrect normalization is more likely to increase than reduce the variation, hence increasing the chance of false discovery, but not that of missing the true differential genes. Based on the high concordance between *cdev* and fold-change MSE on simulation data (Fig. 4), one may expect that similar results would be observed with fold-change MSE. Our analysis indeed confirmed that (Fig. S6).

Real data sets are more challenging, for none of the normalizers could stabilize the true references properly (Fig. S3). Figure 8 shows such results on the Rat Bodymap data sets as an example. In this case, the *cdev* between the raw count matrices and their corresponding ground truths are even smaller than those produced by the normalizers, suggesting that normalization may induce more variation, driving the expression profiles to deviate further from the true patterns. These results imply that on such data sets, no normalization may be better than doing with the wrong assumption.

## DISCUSSION

Although we advocate for ground-truth based measures, we are aware that such preference over data-driven metrics might be debated (*Sun & Zhu, 2012*). Most researchers may choose to combine both ground-truth based and data-driven metrics for a comprehensive assessment. However, we should be cautious that many data-driven criteria commonly used in the field are not justified. Many of these criteria may have arisen from the requirements of statistical analyses and are then mistaken to be the characteristics of the true signals. For one example, matching distribution of read counts is sometimes required by statistical tests. Although quantile normalization could satisfy this requirement perfectly, it would destroy meaningful co-variance structures in the data (*Qiu, Wu & Hu, 2013*; *Qin et al., 2013*). For another example, the reduction of intra- relative to inter-condition variation will improve the statistical significance of variance-based hypothesis testing. A series of transformations were proposed to help equalize the intra-condition variation (*Willems, Leyns & Vandesompele, 2008*), but notably, they were meant to be done after, not in place of, a successful normalization. These examples also illustrate how important it is to separate the assessment of a normalizer from that of any downstream analytic method. These methods sometimes require data transformation, such as logarithm, mean centering, autoscaling, etc. to perform better. Although such operations may improve the outcome, their effects are specific for the analytic workflow. Meanwhile, normalization should bring an expression pattern closer to what actually happens in biological systems, expectedly improving the insights from all downstream analyses.

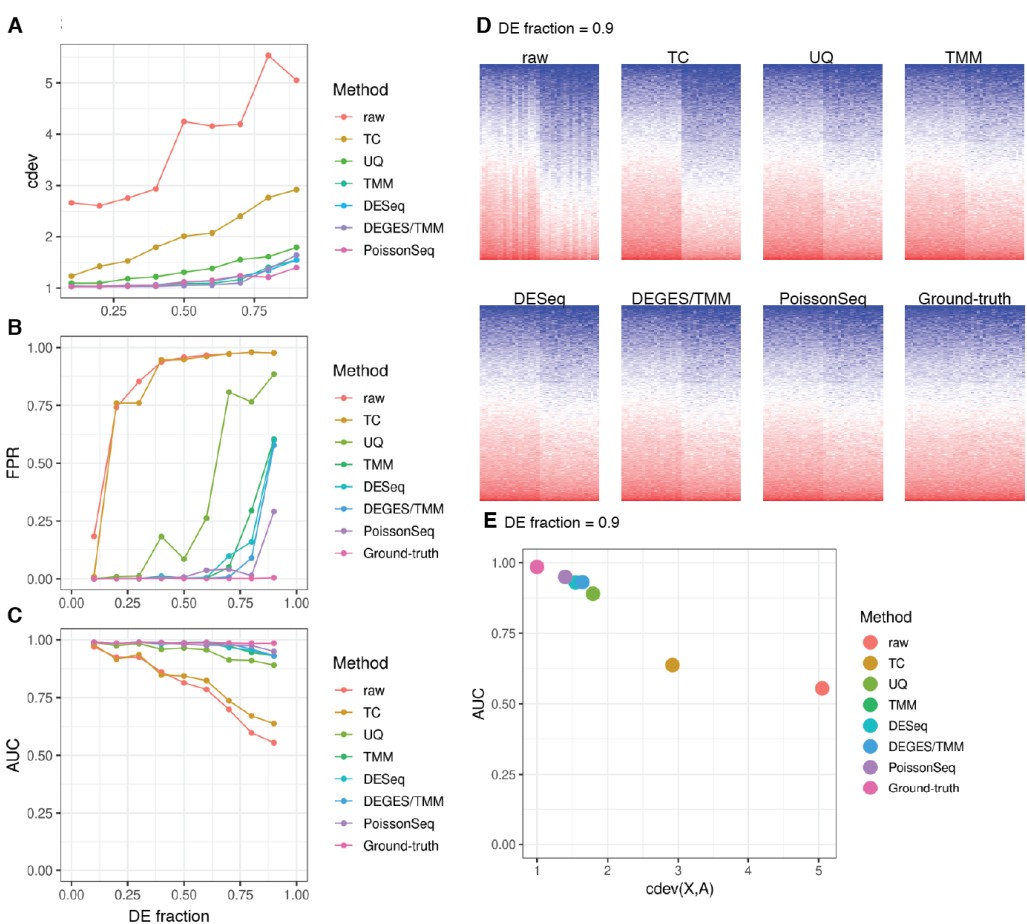

**Figure 7 Application of *cdev* to evaluate common normalization methods on realistic simulations, at varying fraction of DE genes.** (A) Performance of various normalizers. (B, C) DE detection performance as measured by false positive rate (FPR) and area under the ROC curve (AUC). (D) Heatmaps showing expression levels of true references as normalized by different methods, in a simulation with 90% of the genes being differential. (E) AUC of DE detection as a function of *cdev*, on the simulation with 90% of the genes being differential.

Finding a reliable source of ground truth is a significant challenge impeding the use of ground-truth based measures in RNA-seq normalization. Other expression profiling platforms, such as qRT-PCR and microarray, are sometimes resorted to as the means to validate RNA-seq measurements. It goes against common sense to use old technologies to validate newer ones. More formally, *Sun & Zhu (2012)* argued that these platforms are also subjected to measurement errors, thus should not be treated as gold standards. Another way to establish the ground truth is to spike RNA-seq samples with a constant amount of synthetic RNAs such as those devised by the External RNA Controls Consortium (ERCC) (*Jiang et al., 2011*; *SEQC/MAQC-III Consortium et al., 2014*). The usage of ERCC spike-ins for normalization ground truth may also be debated. As observed in previous studies, quantification of ERCC spike-ins was affected by the experimental protocol to extract RNA and prepare sequencing libraries (*Qing et al., 2013*; *Owens et al., 2016*). These

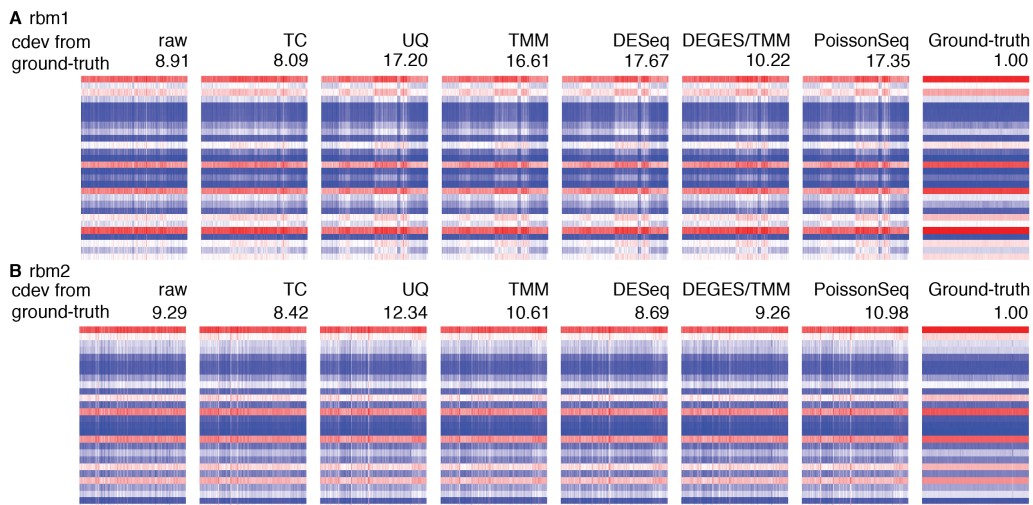

**Figure 8 Application of *cdev* to evaluate common normalization methods on Rat Bodymap data sets.** The heatmaps show expression levels of ERCC spike-ins before and after normalization by various methods, denoted along with their condition-number based deviation from the ground truth.

biases probably happened because the poly-A tails on these synthetic sequences were not optimized for poly-A selection (*Lee et al., 2016*). In general, the depletion of rRNA is recommended over the selection of mRNA as an enrichment protocol, especially in the presence of ERCC spike-ins. However, we made use of RNA-seq assays from libraries prepared by both protocols by separating the samples into individual sets such that those in the same set are prepared by the same protocol. Beside the effects from RNA extraction protocol, ERCC spike-ins sometimes caused concern regarding the wide fluctuation in their fraction of mapped reads across samples (*Qing et al., 2013*; *Risso et al., 2014a*). However, this quantity is equivalent to the proportional abundance, thus are meant to vary with the composition of RNA species. Such compositional variation can be illustrated by a hypothetical experiment in which the samples of conditions # 1 and # 2 are composed of only two genes *A* and *B*. Suppose gene *A* is stably expressed at level 1 and gene *B* is differentially expressed at levels 1 and 3, the proportional abundance of gene *A* is then 0.5 and 0.25 in condition # 1 and condition # 2, respectively. In fact, it is the same reason why normalization is needed for between-sample comparison of gene profiles. A quick simulation revealed that these fractions may differ by two times or more (Fig. S1), similar to what was noted on real data sets by *Qing et al. (2013)*. In light of this, we want to emphasize that the "fluctuation" in fraction of reads mapped to ERCC spike-ins is an expected rather than a concerning behavior.

Once a ground truth is defined, performance can be measured, conveniently and objectively, by a similarity (or dissimilarity) measure. Although MSE is an intuitive and easy-to-compute metric, the definition of fold-change MSE on more than two conditions becomes overly cumbersome. An experiment with $n$ conditions may have $n(n-1)/2$ pairwise comparisons to make, hence the number of fold-change MSE to compute. In contrast, *cdev* is grouping-agnostic and capable of comparing expression matrices as a

whole. As demonstrated in the Results section, *cdev* is also more robust than fold-change MSE, especially on heterogeneous conditions.

Although *cdev* could in principal be computed for any pair of expression matrices $X$ and $Y$, it is not as meaningful outside the context of global scaling normalization. Since $cdev(X, Y)$ is only symmetric when the transformation from $X$ to $Y$ (and $Y$ to $X$) involves only per-sample scaling factors, operations other than that will require additional convention about the direction. However, we would advise against such convention and recommend to keep *cdev* limited to the evaluation of global-scaling normalization. It is a necessary restriction, for these operations are not necessarily mutually exclusive with other non global-scaling ones, such as the removal of batch effects (*Risso et al., 2014a*; *Leek, 2014*), or per-gene normalization (*Li et al., 2017*). For example, global scaling prior to batch effect removal improves biological insight, compared to no scaling (*Tran et al., 2020*). Due to such complementary nature, these operations are not meant to be compared against one another, hence should be assessed by different means.

## CONCLUSIONS

Performance measure has long been in the shadow of RNA-seq normalization, despite a vibrant scene of normalizing method development. Most of the *de facto* measures are either qualitative or potentially subjected to biases. Some were introduced without much justification. We elaborated on the definition of *cdev*, its properties and applications, showing that this metric is highly indicative for normalization performance, proportional to both upstream processing and downstream analyses in simulations. Along with this metric, we re-processed and curated a collection of RNA-seq data sets with ERCC spike-ins. We envision this toolset will be highly valuable in developing and evaluating normalization methods.

## ACKNOWLEDGEMENTS

The support and resources from the Center for High Performance Computing at the University of Utah are gratefully acknowledged. We thank Dr. Orly Alter for feedback on the manuscript.

### Funding

This work has been supported by the National Science Foundation (CAREER grant 1350344 to Matthew Might). The funders had no role in study design, data collection and analysis, decision to publish, or preparation of the manuscript.

### Grant Disclosures

The following grant information was disclosed by the authors:
The National Science Foundation: CAREER grant 1350344.

## Competing Interests

The authors declare there are no competing interests.

## Author Contributions

- Diem-Trang Tran conceived and designed the experiments, performed the experiments, analyzed the data, prepared figures and/or tables, authored or reviewed drafts of the paper, and approved the final draft.
- Matthew Might analyzed the data, authored or reviewed drafts of the paper, and approved the final draft.

## Data Availability

Source codes for all the analyses in this paper are available on GitHub, at https://github.com/ttdtrang/cdev-paper.

R data packages are available on GitHub, at https://github.com/ttdtrang/spiked-rnaseq-collection.

## Supplemental Information

Supplemental information for this article can be found online at http://dx.doi.org/10.7717/peerj.12233#supplemental-information.

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
