# Peer review of "cdev: a ground-truth based measure to evaluate RNA-seq normalization performance"

_PeerJ, doi:10.7717/peerj.12233_

## Round 0.1 · original submission · Minor Revisions

The reviewers have a number of suggestions that may be imlemented to improve the paper. While some of them require additional calculations, the reviewers consider them optional, and the majority of comments relate to the presentation rather than the substance.

Reviewer 1 ·

Basic reporting

The manuscript is well-written and has a clear structure. Unfortunately, I could not find the GitHub repo and check that the data and code are available, although stated in the main text l. 153 (see "Validity of the findings" comments where I debate for the code’s availability and checking it up before publication). I apologize if it is my fault and the link is already present in the manuscript. I recommend making it less obscure for the reader then.

Although I find the main text figures essential and relevant for understanding the main text, I could not say so about Figures 8B, D. I find data representation in these figures redundant and its purpose unclear. These two figures demonstrate the matrices of pairwise cdev values between different matrices after normalization.
These matrices have ones on the main diagonal, as expected from the properties of cdev provided by the authors in l. 198 (cdev(X,X')=1 when X' is a scaled version of X; and on the main diagonal X'=X).
The matrix is symmetrical, as should be according to the symmetry property of cdev (l.192).
Thus the main diagonal and lower triangle can be excluded from the visualization without the loss of significant information to the reader.
Finally, the values of cdev between different normalizers are not discussed in the text. It is not clear what conclusions can be made from presenting this data. I suggest the authors retain the first row of matrices 8B and 8D in the main text (with the exclusion of cdev(raw, raw), and rather place the full versions to Supplementary materials. Alternatively, the full matrices’ results can be explained in the main text if they are indeed relevant.

Few unclearly defined terms make the reader stumble and look for additional literature or other clues in the text:
- n and m in l. 155 presumably represent the number of genes and the number of samples, correspondingly. This is not stated in the text, though.
- "Oracle" (a title used for ground-truth normalization in the main figures) is not defined in the main text.
- For the simulations’ description, the authors use the terms "simplistic" and "realistic.” This corresponds to "single" versus "continuous" fold change in Table 1, correspondingly. However, to my understanding of Figure 2, these two regimes are also named "homogeneous" versus "heterogeneous" fold change. Intuitively, this seems natural renaming of these regimes. However, this confuses the reader as it also sounds like the introduction of the new entity in simulations. I suggest renaming the titles in Figure 2 and Table 1 to be the same.

There are few typos that can be easily checked for and corrected. I provide what I noticed:
- Figure 2 title "realistic" instead of "realistic"
- l. 381 "that will requires"
- l. 120-122 "we elaborate", but "We demonstrated" in the next sentence.

Experimental design

The paper's experimental setup allows for comprehensible and unbiased testing of both cdev performances as the measure of normalization quality and performance of different non-ground truth normalizers. However, because both of these aims are strived for, the authors' debate is not completely persuasive.

For example, in Figures 4, 5 the authors demonstrate the number of tests on simulated and real data that demonstrate that cdev sometimes produces results similar or better to the traditional FC MSE approach on ground-truth normalizations. Next, for testing different normalizers, the authors use only cdev as they assume its benefits proved.
Indeed, the authors demonstrate that there are cases when FC MSE does not detect the bad normalization (Figure 6), and cdev has great discriminative power. However, the authors do not describe the opposite cases clear from Figure 6A and Figure 5 "Heart-Lung" comparison, when cdev is closer to 1. At the same time, FC MSE is large and clearly demonstrates the low quality of normalization.
To solve this conundrum, I would suggest one of two (or both) improvements:
1. reporting a close-up examination of one opposite case in Supplementary materials (e.g., demonstration of normalization and expression heatmap before and after normalization for the case with FC MSE~1 and cdev ~4 in "Heart-Kidney" dataset);
2. providing the FC MSE alongside Figures 7 and 8 in Supplementary materials so that the reader can assess all normalizers' performance in an unbiased manner.

As a researcher who will potentially use cdev to assess normalization purposes in my experiments, I find it hard to judge when the normalization is good or not. In Figure 7A, one can observe that number of DE errors increases with larger cdev, but is there a threshold above which I should consider my normalization as "bad"?
I believe that the authors have all the power to take one step further toward this naturally arising "real-life" problem. I suggest the authors can propose the threshold value of cdev given the number of total and DE genes that will notify the researcher whether the normalization was "good" or "not".

Finally, if I do not have extensive computation powers to calculate cdev (as cautioned by the authors in l. 377-378), and I have access only to simplistic FC MSE, how can I rely on the results of this study to decide on the success of normalization? If not introducing the same threshold definition for MC MSE, I would suggest providing Figure 2 with FC MSE instead of cdev in Supplementary materials.

The author's selection of normalizers is made clear in l. 207: "These methods are chosen due to the simplicity of computation and/or availability of R implementation, which allows quick integration to the current analysis." However, the selection does not include the best existing normalizer called graph-based normalization (proposed by the authors earlier, Tran et al. 2020), probably due to its computational complexity. Nevertheless, I find it important (although optional) to add this normalizer and test cdev not only on mediocre normalizers but also for a range of real datasets with the best normalizer known to the literature (although only part of datasets can probably be processed with gbnorm).

The last unclear point on experimental design, can the authors discuss why the uniform distribution of fold change values was selected for simulations? Is it close to the distribution observed in real datasets?

Validity of the findings

The authors test a cdev method to assess the quality of normalization, which was proposed by the same team earlier (Tran et al. 2020). The usage of the already-published method debates for its technical validity, although I noticed two potential weak points.

First, when cdev is computed, it uses a conditional number of Moore-Penrose pseudoinverse multiplied with the expression matrix, or kappa. In R, there are several parameters of the kappa function that control the properties of calculation [1], including whether the exact or approximate result is returned. If the approximate calculation was used, can the authors guarantee its adequacy and numerical stability?
To my experience, the calculation of the smallest singular values may be poorly implemented in some programming languages and result in inadequate results. If the cdev code was made available, I would try to validate the kappa method on a number of datasets and check its numerical stability.

1. https://stat.ethz.ch/R-manual/R-devel/library/base/html/kappa.html

Second, cdev relies on the assumption that the normalization is agnostic to the scaling (l. 177-179). However, to my understanding, some DE calling algorithms rely on the absolute count of the gene expression to derive the statistical significance of expression changes (e.g., DESeq Anders & Huber 2010).
As I am not a specialist in limma toolkit used by the authors, can they comment whether it is not the same for calling DE genes with limma?
Also, I suggest discussing this problem when explaining the principles of cdev, since a perfectly normalized dataset by cdev criterion might require additional scaling to produce valid results with such a popular tool as DEseq.

Additional comments

I find the research an important contribution to the field, an extensive benchmark of two normalization criteria (cdev and FC MSE) for several normalizers and various simulated and real datasets. Although both testings against ground-truth with ERCC spike-ins (Tran et al. 2020), cdev, and FC MSE were introduced before, there is a clear need for a study like this one will describe the properties and behavior of these entities on a set of simulated and real-life datasets. Although not new, this paper's theoretical contribution is clear, as it provides the strict formulation and practical demonstration of cdev.
In my opinion, the paper lacks few improvements and additional validations before the publication.

·

Basic reporting

0) l171-202: description of sdev is no clear. Authors introduce expression matrix as n*m matrix where n is number of genes and m is number of samples (l155). Then they define matrix B as XB=A where X and A are expression matrices of the same size. But on line 192 matrix B is n*n matrix while to make XB=A correct B should be m*m. Maybe authors use same variable in different places for different porpoises, but in this case it is still unclear whether number of rows (columns) in B is equal to number of genes or samples. From the text it look like latter is correct, if so B should be not a solution of XB=A but approximation, but it is not discussed. Authors suppose that B is diagonal, but in reality it obviously never will be diagonal at least due to random noise.
1) Authors should provide access to code: functions to compute sdev and examples of it application. Authors say that all code is available on GitHub but provide no links. It is very important taking into account uncertainties in description of the method.
2) l213: "The resulted log-cpm matrix was then passed to limma functions, along with the per-sample scaling factors computed according to each method." - sclaing factors usually used under cpm calulation (and cpms usually already scaled for them). So. mostl likely scaling factors were not supplied separately from cpms, please clarify
3) It is commonly assumed that count-based methods such as edgeR or DESeq are prefered under methods based on normal distributions such as limma. While I do not think that it affect result too much I would suggest to use more convenient methods or at least explain why limma was used.
4) S2: x-axis show samples and different spike-ins are shown by lines, right? I suggest to add this information into legend. Figure title is misleading because it shows read counts rather than correlation, one can show distribution of all correlation coeeficients as well. I would suggest to substract mean spike-in expression at least from dmelAC dataset to make agreement more visible
5) It is completely unclear what is shown on fig 2. Authors should explicitly describe the procedure. What "fraction of differential genes in scaling" stands for?

Experimental design

authors use proper experimental design

Validity of the findings

taking into account incomplete description of the methods it is hard to conclude on validity

Additional comments

Authors introduce new method sdev to compare two expression matrices for evaluation of different normalization procedures. I cannot see applications for the method in majority of RNA-Seq experiments because most of them do not use spike-ins that can be used to check normalization quality. But maybe sdev can be used to develop new normalization procedures while I'm not sure that they are needed. In any case currently it is hard to conclude on manuscript quality because source code is not available and methods description is not clear

Reviewer 3 ·

Basic reporting

The authors use clear, unambiguous and professional English in their paper. The background of the problem is well explained and supported with relevant literature references.
The structure of the article is clear and comprehensive. Provided tables and figures sufficiently illustrate statements authors make in manuscript.
Authors did not provide any links to code or generated counts data they used for the analysis.

Experimental design

The focus of the research is evaluating different normalization methods for RNA-seq based data, which is an important question in the field of Differential Gene Expression. It fits well the scope of the journal. Authors suggest a method that they claim outperforms existing popular normalization approaches.
Authors have used acceptable number of datasets differing in groups heterogeneity and effect size. Sufficient number of state-of-art normalization techniques were employed for the comparison.
Description of applied methods lacks detail and does not give sufficient information for one to reproduce the analysis. The main theoretical concept authors propose, in turn, is described thoroughly and in detail.

Validity of the findings

Authors compiled several RNA-seq experiments with ERCC spike-ins from several organisms, of different size, structure and with addition if different spike-in mixtures. On real data sets, it was shown that cdev shows results following the theoretical predictions. Also, based on the comparison of cdev results with log-fold MSE, authors claim that it is a more robust alternative to the latter one.
Conclusions of the paper are well-supported by the provided analysis results.

Additional comments

I suggest several things that can improve the work.
Counts data and code used for the analysis should become accessible, and more detail of the analysis to the methods section should be added.

---

## Round 0.2 · Major Revisions

Please address the comment that the real case procedure, as opposed to the general one, can be simplified and made much more efficient.

Reviewer 1 ·

Basic reporting

The authors resolved all the comments.
Like an overlay of page number and the figure caption (p.5), some stylistic problems should be double-checked. Also, the GitHub links still cannot be found in the text. Clearly, it's a problem with either compilation of the file or the submission system, as the authors have readily provided the links with the rebuttal letter. This can be double-checked with the editor or support staff.

Experimental design

The authors resolved all the comments.

Validity of the findings

The authors resolved all the comments.

Additional comments

All the concerns were thoroughly investigated and clarified. I believe that the manuscript is ready for publication with PeerJ.

·

Basic reporting

while in general the paper is clearly written and easy to follow it seems to over-complicates quite simple idea (see below). Under used assumptions proposed method can be described in a much simpler way (ratio of maximal and minimal normalization coefficients) and implemented much efficiently (see attached R markdown).

Experimental design

study design is appropriate

Validity of the findings

findings are valid

Additional comments

Thanks to provided code it is now clear how cdev works. In general authors consider family of read count matrix normalization procedures that can be expressed as XB=A where X and A are read count matrices and B is m*m normalization matrix where m is number of samples. So, this family of normalization methods include all possible linear transformation but do not include more advanced approaches that uses different normalization coefficients for different genes (as SCT from Seurat for instance). As far as I know out of the whole considered family of normalization methods only simple per-sample scaling (that corresponds to the diagonal B) is used in practice, and it is the only case considered by authors, most important properties of cdev take place only under this assumption. So, only diagonal B are reasonable and only them are considered in the paper. Diagonal matrix is easy to consider as a vector of normalization coefficients. Since singular values of diagonal matrix are equal to its diagonal elements, cdev is simply a ratio of maximal and minimal diagonal element (normalization coefficient). In attached R markdown I show that one-row function can calculate cdev 500-time faster, than method proposed by authors.
While cdev is rather trivial method, it still might be useful, the paper clearly show that it correlates with other measurements of normalization quality. But current description of the method is foolishly over-complicated and implementation is obviously inefficient.

---

## Round 0.3 · Minor Revisions

The reviewer has commented that the general case might be downplayed, but left it to the authors to decide; I concur and I'll be willing to accept the paper as is. Still, I make a formal decision of minor revision to retain a possibility for rewriting (without additional reviewing cycle). If the authors decide not to, they may resubmit the same text.

·

Basic reporting

all my comments are addressed in the revision

Experimental design

design is correct

Validity of the findings

findings are valid

Additional comments

I still think that general case of sdev is emphasized too much in the paper taking into account that only special case of diagonal B is considered. But since it doesn't affect validity of the paper i think that whether this part should be shortened or not must be decided by editor or/and authors.

---

## Round 0.4 · accepted · Accept

As I've left the decision on rearranging the manuscript with the authors, the present version is acceptable.